# Disadvantages in preparing and publishing scientific papers caused by the dominance of the English language in science: The case of Colombian researchers in biological sciences

Valeria Ramírez-Castañeda[1,2,3]*

1 Department of Catalan Philology, University of Barcelona, Barcelona, Spain, 2 Department of Integrative Biology, University of California, Berkeley, California, United States of America, 3 Museum of Vertebrate Zoology, University of California, Berkeley, California, United States of America

* vramirezc@berkeley.edu

## Abstract

The success of a scientist depends on their production of scientific papers and the impact factor of the journal in which they publish. Because most major scientific journals are published in English, success is related to publishing in this language. Currently, 98% of publications in science are written in English, including researchers from English as a Foreign Language (EFL) countries. Colombia is among the countries with the lowest English proficiency in the world. Thus, understanding the disadvantages that Colombians face in publishing is crucial to reducing global inequality in science. This paper quantifies the disadvantages that result from the language hegemony in scientific publishing by examining the additional costs that communicating in English creates in the production of articles. It was identified that more than 90% of the scientific articles published by Colombian researchers are in English, and that publishing in a second language creates additional financial costs to Colombian doctoral students and results in problems with reading comprehension, writing ease and time, and anxiety. Rejection or revision of their articles because of the English grammar was reported by 43.5% of the doctoral students, and 33% elected not to attend international conferences and meetings due to the mandatory use of English in oral presentations. Finally, among the translation/editing services reviewed, the cost per article is between one-quarter and one-half of a doctoral monthly salary in Colombia. Of particular note, we identified a positive correlation between English proficiency and higher socioeconomic origin of the researcher. Overall, this study exhibits the negative consequences of hegemony of English that preserves the global gap in science. Although having a common language is important for science communication, generating multilinguistic alternatives would promote diversity while conserving a communication channel. Such an effort should come from different actors and should not fall solely on EFL researchers.

**Data Availability Statement:** All relevant data are within the manuscript and its Supporting

Information files. Including complete survey questions and results.

**Funding:** The author(s) received no specific funding for this work.

**Competing interests:** The authors have declared that no competing interests exist.

## Introduction

At the same time that scientific articles became the measure of scientific productivity, English was imposed as the language of science, culture, and the global economy [1]. As a consequence, today 98% of publications in science are written in English, especially in the areas of natural and basic sciences, establishing English as the *lingua franca* of science [1]. This creates a disadvantage for scientists with English as a Foreign Language (EFL) because they must publish complex texts in a foreign language to advance their careers [2]. This disadvantage gives rise to global inequalities, especially in countries where the majority of the population receives minimal English training and bilingualism with English is very low [3]. Thus, English proficiency and socioeconomic level influence scientific success, access to knowledge and expatriation, among others.

One of the most important goals for modern society is to increase scientific production from Africa, Latin America, Middle East, and developing Asia. There is a strong correlation among English proficiency, economic development, and technological innovation in terms of number of articles, number of researchers and research and development expenditure [4]. Therefore, the prevalence of the English language in the sciences deepens the inequality in knowledge production between countries with high and low English proficiency [5], maintaining the gap in scientific production between the countries of the global south or peripheral and the countries of the global north (include the G8 countries and Australia), reducing the individual scientific contributions of EFL scientists [6]. Together these factors limit the advancement of the broad scientific communities within those countries [7].

Numerous studies have identified the use of English in academia as a source of inequality and segregation in science [8–12]. These inequities affect the scientific community at multiple levels. In local communities of EFL countries, scientific thinking is harmed, particularly in higher education, as learning depends on cultural attitudes derived from the native language spoken by the students, and science becomes alien to their own experiences [13–15]. Diversity in language promotes diversity in thinking, affecting creative process and imagination; thus, the maintenance of multilingualism in science could have an impact on scientific knowledge in itself [14].

Local journals are a refuge for communication of scientific research in languages other than English, nevertheless they are often perceived as low-quality, since the most important research work is often reserved for international journals. Therefore, readers with language barriers only have access to limited studies that the researchers consider not complete, important or broad enough to be published in an international journal. Local readers often are unaware of the most significant research being conducted in their region, which has resulted in a void in information important for political decision making, environmental policies, and conservation strategies [16–18]. In addition, despite the importance of local knowledge, the professional success of a scientist correlates to a greater extent with their "internationalization". This constant pressure could be influencing academic migration, known as "brain drain". English learning is one of the pressure factors of migration, as it is more difficult to achieve upper English proficiency for scientists who remain in EFL countries [15, 19, 20].

In periphery countries there is a strong relationship between English proficiency and socioeconomic origin, thus it is important to understand the publishing costs associated with the socioeconomic origin of researchers. Among Latin America, Colombia is the second most unequal territory: in 2018 it invested only 0.24% of its GDP (Sweden investment was 2.74% of its GDP) in science, technology and innovation [21], and it has one of the lowest levels of English proficiency among the world rankings [4]. In addition, for 2019, Colombia had only 58 researchers per million inhabitants [22, 23]. This study aims to determine if Colombian doctoral students of natural sciences face disadvantages when publishing scientific articles in

English, compared to publications in their first language, and to quantify the extra work that these scientists put into writing, reading, and presenting their work in English. In addition, this study examines the impact of socioeconomic background on English proficiency and the costs it generates when publishing.

## Materials and methods

In order to determine the costs of publishing in English experienced by Colombian researchers in biological sciences, 49 to academics were surveyed. These researchers completed their PhDs or are enrolled in doctoral studies and are attempting to publish. They participated in the "Implications of language in scientific publications" survey containing 44 questions in Spanish language (S3 and S4 Files). This survey was available for two months and shared directly to researchers and on Twitter under the hashtag "#CienciaCriolla" (used between Colombian researchers). Responses were anonymous. It must be mention that the researcher's demography in Colombia is gender, ethnic, and socioeconomic biased. Only 30.21% of natural science researchers are women [24], researchers come primarily from big cities [25], and undergrad students come mainly from middle and high socioeconomic classes [26]. Therefore, it would not have been possible to completely control for bias in who took the survey. It must be also recognized that without specific numbers for total Colombian researchers in biological sciences, 49 may not be a representative sample size from which to draw accurate statistical inferences.

Additionally, the prices offered by prestigious scientific publishers for translation (Spanish to English) and editing of scientific texts were searched to measure the economic impact in relation to a Ph.D. student salary in Colombia [27–31].

## Survey construction

The main survey of this work, entitled "Implications of language in scientific publications," has 44 questions divided into three sections: basic data, writing articles in English, and learning English (S3 and S4 Files). This survey sought for the most quantitative approach as possible, however, each question is inevitably under some degree subjectivity due to human interpretation. The responses obtained were grouped for statistical analysis (Table 1).

## Statistical analysis

Statistical analyses were performed in R v.3.6.1 [35] and data were plotted with the ggplot package [36]. To compare reading and writing between English and Spanish, time investment and the level of anxiety in conferences participation, an ANOVA was performed (*aov* in package 'stats' v3.5.3). The margin of error was calculated with 95% confidence. An Analysis of Principal Components (PCA) was performed using the variables contained in the "English proficiency" and "Socioeconomic data" groups for reducing redundancy in the variables (*PCA* in package FactorMiner v2.2). The proportion of variance explained by each principal component was reviewed, and only the first principal component was retained for each dataset, as it described 51% and 62% correspondingly of the total variation. Subsequently, a linear regression was executed with the intention of comparing these two variables, English proficiency PC1 vs socioeconomic status PC1 (using *lm* in package 'stats' v3.5.3).

## Editing and translation service costs

In order to visualize the prices of English editing and translation services for scientific texts, information was sought in five of the most relevant scientific publishers [27–31]. The

**Table 1. Grouping of survey questions in statistical analyses.**

| Group | Variable | Type of variable | Reference |
|---|---|---|---|
| **Socioeconomic data** | Differentiation between attending public and private education in the high school and university | Binary (public/private) | [32] |
| | Occupation of the father and mother to define socioeconomic status | Categorical | [33] |
| | Socioeconomic stratification in Colombia from him- or herself, father, and mother | Ordinal (1 to 6) | [34] |
| **English proficiency** | First language(s) | Categorical | |
| | Language of the country in which they live | Categorical | |
| | Proficiency in writing, oral expression, reading and listening in English | Ordinal (bad, regular, good, excellent) | |
| | Age of beginning of English learning | Discrete quantitative | |
| | Years studying English | Discrete quantitative | |
| | Satisfactory experience in the English learning institution | Ordinal (1 to 5) | |
| | Years lived in English-speaking country | Discrete quantitative | |
| | Percentage of English speaking daily | Percentage quantitative | |
| **Publications** | Number of scientific publications reviewed by academic peers | Discrete quantitative | |
| | Percentage of those publications in English or in a language other than English | Percentage quantitative | |
| **Costs of writing in English** | Payment or favors for editing or translating scientific publications | Binary (yes/no) | |
| | Percentage of articles—payment or favors for editing or translating scientific publications | Percentage quantitative | |
| | Number of rejections in scientific journals associated to English writing | Discrete quantitative | |
| | Time spent writing a scientific article in English or Spanish (labor hours) | Continuous quantitative | |
| | Preference between writing directly in English or translating to English | Binary (English/translating) | |
| **Writing difficulties in English or Spanish** | Difficulty of writing scientific paper introduction, methods, results, discussion, conclusion | Ordinal (1 to 5) | |
| **Reading difficulties in English or Spanish** | Difficulty in understanding general text / Understanding scientific terminology / Interpreting figures and tables | Ordinal (1 to 5) | |
| **Participation in conferences** | Oral presentations in international conferences | Binary (yes/no) | |
| | Level of anxiety in oral presentations in English or Spanish | Ordinal (1 to 5) | |

information and costs of these services are public and can be obtained through the web pages of publishers. All data were taken with respect to prices for a text of 3000 words, as that is the average length of a scientific article; searches were performed in October 2018.

These publishers offer two types of editing services, a three-day service (premium) and a one-week service (standard); both prices were used for the analysis. Only the prices for Spanish—English translations were used. Finally, these prices were compared with an average doctoral salary in Colombia [25], 947 US dollars or 3 million Colombian pesos (1 US dollar = 3.166 Colombian pesos, exchange price on January 31, 2019).

## Results

A total of 49 responses were obtained from Colombian doctoral students or doctorates in biological sciences whose first language is Spanish. From Colombians' surveyed 92% (sd = 0.272) of their published scientific articles are in English and only 4% (sd = 0.2) of their publications were in Spanish or Portuguese. In addition, 43.5% of the doctoral students stated at least one rejection or revision of their articles because of the English grammar.

With regards to time investment, there was a significant increase in the time invested writing a scientific article in English in comparison to Spanish for survey participants (Fig 1). The process of writing in Spanish takes on average 114.57 (sd = 87.77) labor hours, while in English, 211.4 (sd = 182.6) labor hours. On average, these scientists spend 96.86 labor hours

more writing in English. However, 81.2% of the doctoral students stated that they prefer to write directly in English in comparison to writing in Spanish and then translating into English.

The need for editing or translation of scientific texts is widespread among Colombian doctoral students. Among the respondents, 93.9% have asked for favors to edit their English and 32.7% have asked for translation favors. Regarding the use of paid services, 59.2% have paid for editing their articles and 28.6% have paid for a translation.

The *Premium* editing total cost and the standard translation cost represent almost a half of an average doctoral monthly salary in Colombia (Fig 2).

Reading comprehension is also affected by the language of the text (Fig 3). However, only 18% of respondents prefer to read scientific articles in Spanish than in English. On the other hand, neither the interpretation of figures nor the understanding of scientific terminology is affected by the reading language.

To analyze the difficulty of writing scientific articles in two languages, survey participants were also asked how they found it difficult to write different sections of articles: introduction, methods, results, discussion, and conclusions. In all cases, survey participants found the discussion was the most difficult section to write, while the methods were perceived as "easier" (Fig 4). Overall, all sections except methods are perceived as significantly "more difficult" to write in English than in the participant's first language.

With regard to the use of English in oral presentations at international events and conferences, 33% of respondents stated that they have stopped attending due to the mandatory use of English in oral presentations. Additionally, greater anxiety was perceived when presenting papers orally in English than in Spanish (Fig 5).

In order to determine whether or not the socioeconomic origin of doctoral students affects their proficiency in English and in turn increases the costs of publishing in English, an analysis

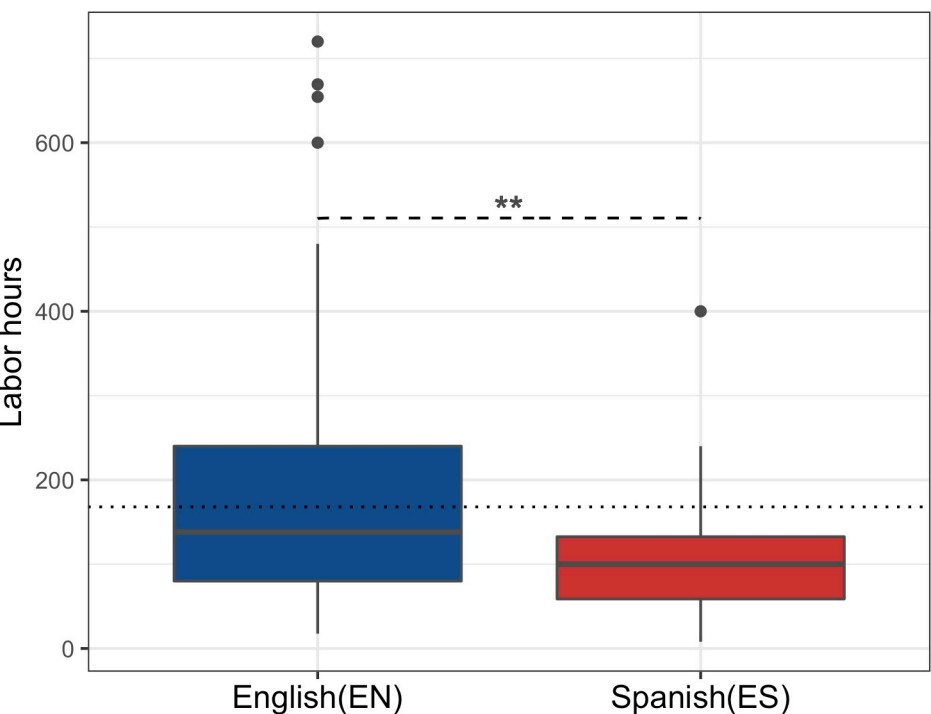

**Fig 1. Time invested writing a scientific article in English (EN) as a secondary language and in Spanish (ES) as a first language.** An ANOVA analysis was performed to compare the variables obtaining an F-value = 7.095 and p-value = 0.00951 **. The dotted line represents labor hours per month.

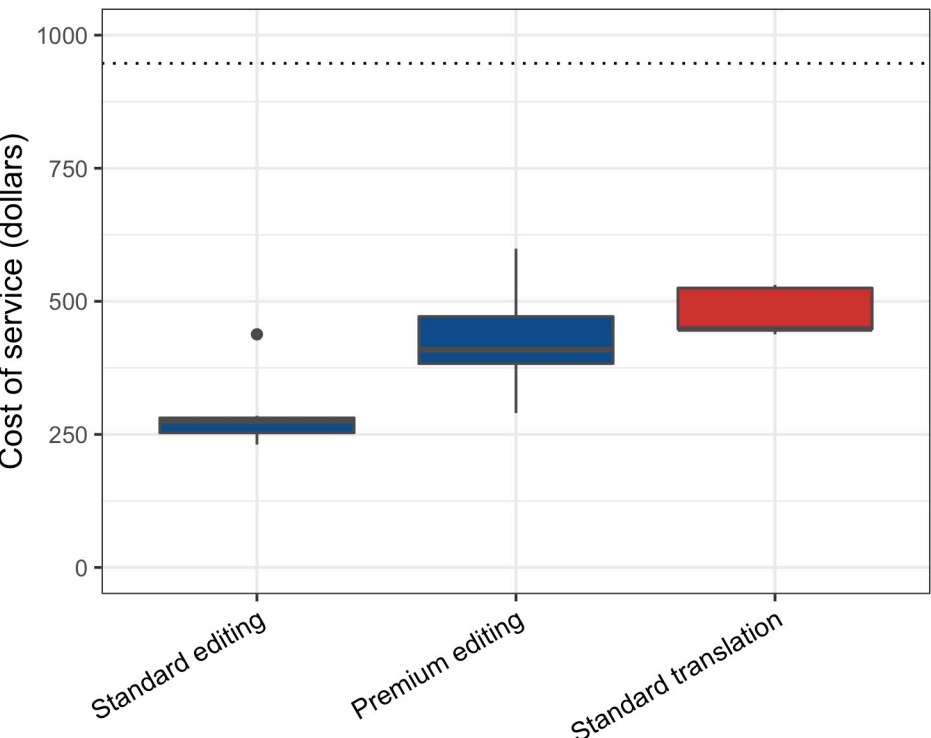

**Fig 2. Cost of translation and editing services of scientific articles from six publishers.** The Y axis is the price of the service in US dollars, the X axis represents the type of service, the standard or premium service corresponds to the delivery days. The dotted line represents an average Ph.D salary in Colombia ($ 947).

of principal components was used reduce survey data related to socioeconomic background or English proficiency into single variables because both represent more than the 50% of the whole variance. For the following analyzes: 1) English proficiency is represented by PC1_English_proficiency, which explains 51% of the variance of the survey variables that are related to this subject (see methods), 2) the socioeconomic status is represented by PC1_Socioeconomic_status, which represents 62% of the variance of the variables of the survey that were related to this denomination (see methods). The socioeconomic status explains 15% of the English proficiency of researchers (Fig 6), which means that family and economic resources are partly translated into more proficient English.

## Discussion

Many of the factors relating to publishing in English assessed in our study represent substantial costs in time, finances, productivity, and anxiety to Colombian researchers. Interestingly, the researchers appear to prefer to read and write articles in English and the scientific terminology do not represent an additional cost for Colombian researchers. In addition, a correlation between the socioeconomic status and English proficiency was found, suggesting an intersectional effect of language in science. These results can be extrapolated to understand costs of the English hegemony to all South American researchers, that in part contributes to a global gap between native English-speaking scientists (NES) and EFL scientists. This gap makes apparent the necessity of recognizing and protecting multilingualism in science. Although having common language is important for science communication, this effort should involve different actors in the research community and not only EFL researchers' effort.

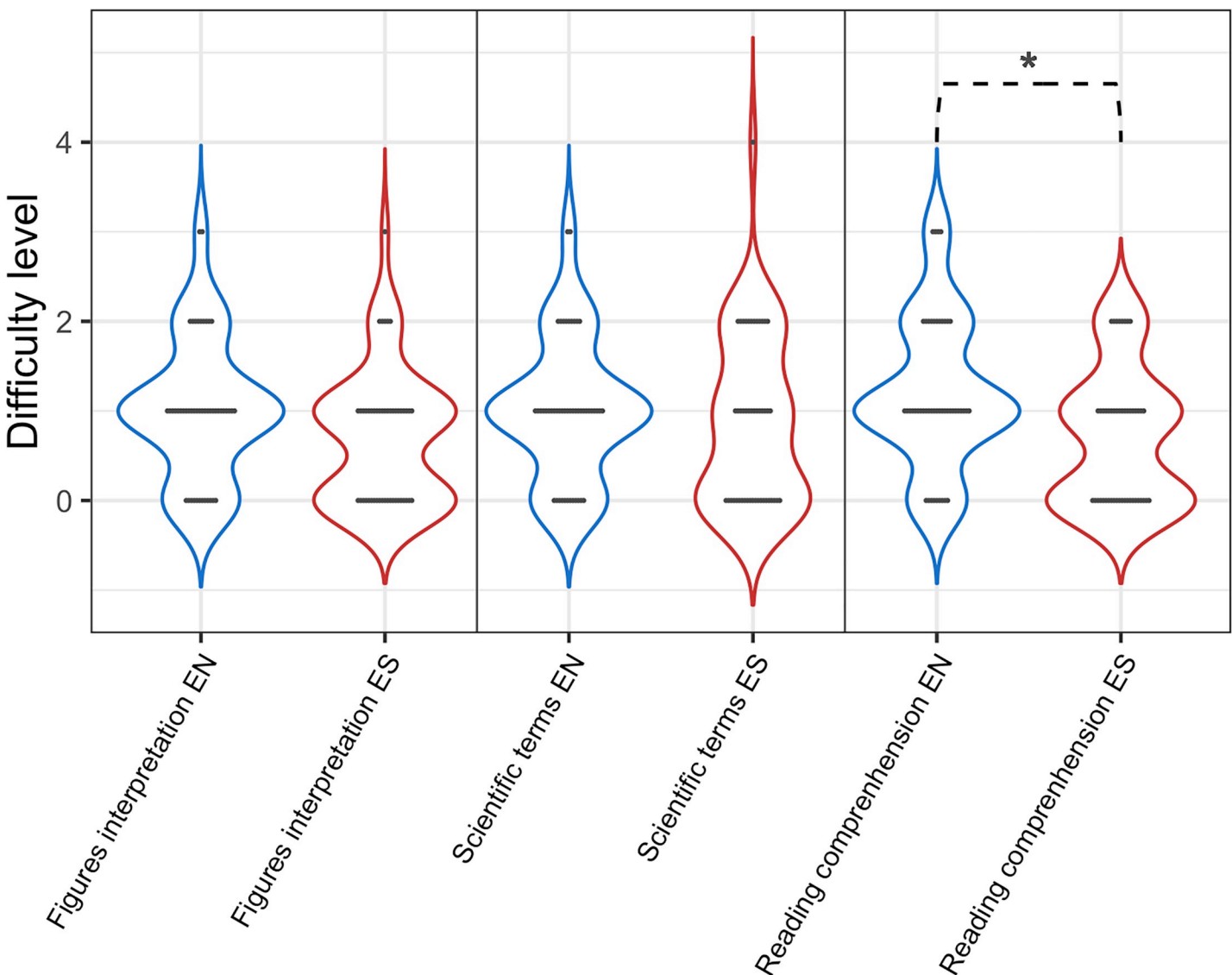

**Fig 3. Reading comprehension, interpretation of figures, scientific terminology in English (EN) vs. Spanish (ES).** A *Poisson* regression was used to analyze these discrete ordinal variants (Qualification from 1 to 5). A Chi-squared test was performed between languages for each category: interpretation of figures (Z-value = 0.756, Pr (Chi) = 0.09754), understanding of scientific terminology (z-value = 0.143, Pr (Chi) = 0.4619) and reading comprehension (z-value = 1.427, Pr (Chi) = 0.01209 *).

Our results show that several factors could lead to disadvantages of EFL researchers. The time investment in writing an article in English, for example, increases on average by 96.86 labor hours. This variable was not directly measured; it is based on the subjective perception of time of each person. However, as Guardiano and collaborators [37] suggest, this extra cost affects the time spent on scientific tasks, decreasing the scientific productivity of researchers. Regarding the economic costs, between 50% and 30% of respondents have hired services to correct or translate scientific texts. To contextualize the cost of these services, a doctoral student should invest one-quarter to one-half doctoral monthly salary per article. It should be taken into account that scholarships and financing opportunities for doctoral students in the country are scarce [38], and not all of them have access to the forgivable loans provided by governmental institutions. More than 90% of researchers have asked for English-editing favors,

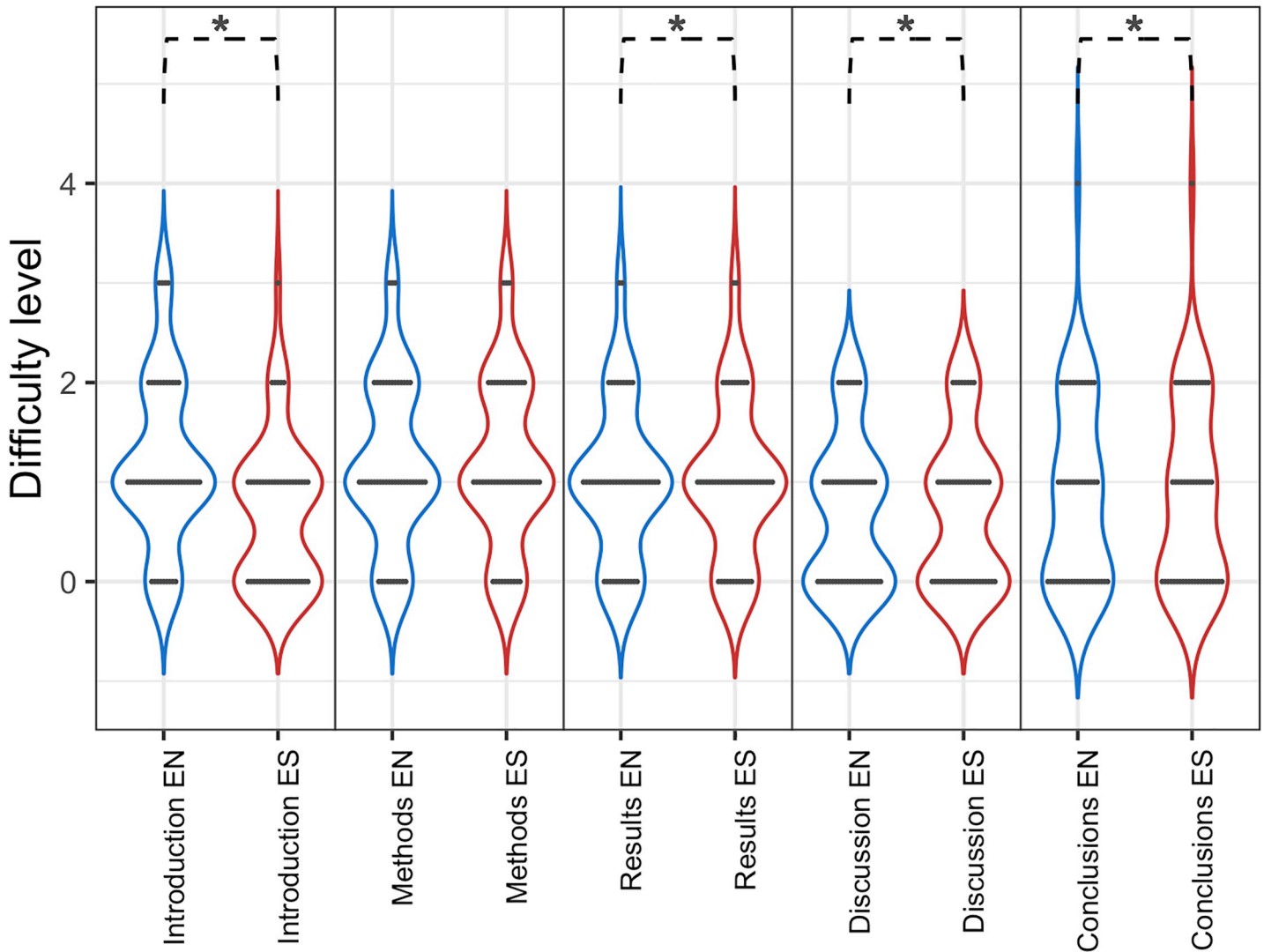

**Fig 4. Difficulty of writing the different sections contained in a scientific article in English (EN) and Spanish (ES).** A *Poisson* regression and Chi-square test was carried out: Introduction (z-value = 9.325, Pr (Chi) = 0.0158 *), methods (z-value = 3.046, Pr (Chi) = 0.07057), results (z-value = 4.899, Pr (Chi) = 0.04397 *), discussion (z-value = 11.732, Pr (Chi) = 0.02384 *), and conclusion (z-value = 7.688, Pr (Chi) = 0.03956 *).

but favors are unpaid labor that may have subsequent costs. The cost of this favor particularly leans on the weakest in the relationship, in this case, the EFL researchers because their career depends on publishing in a second language. Therefore, ensuring a permanent source of "favors" is essential for an EFL researcher that is willing to negotiate for "help" by reinforcing dependence with research groups or scientists in NES countries [8]. Romero-Olivares [39] exemplified this point by showing a reviewer comment "The authors need a native English-speaking co-author to thoroughly revise the grammar of this manuscript.", or as Ordoñez-Matamoros et al [40] mention for Colombian researchers "co-authoring with partners located in foreign countries tend to publish their work in journals of higher impact factor and receive more citations per article than those not co-authoring with partners located overseas".

Around 80% of the respondents prefer to read and write scientific content directly in English. However, this result could be interpretable as "obligation" rather than as "preference" because of the monolingualism of scientific readings and the pressure to publish in

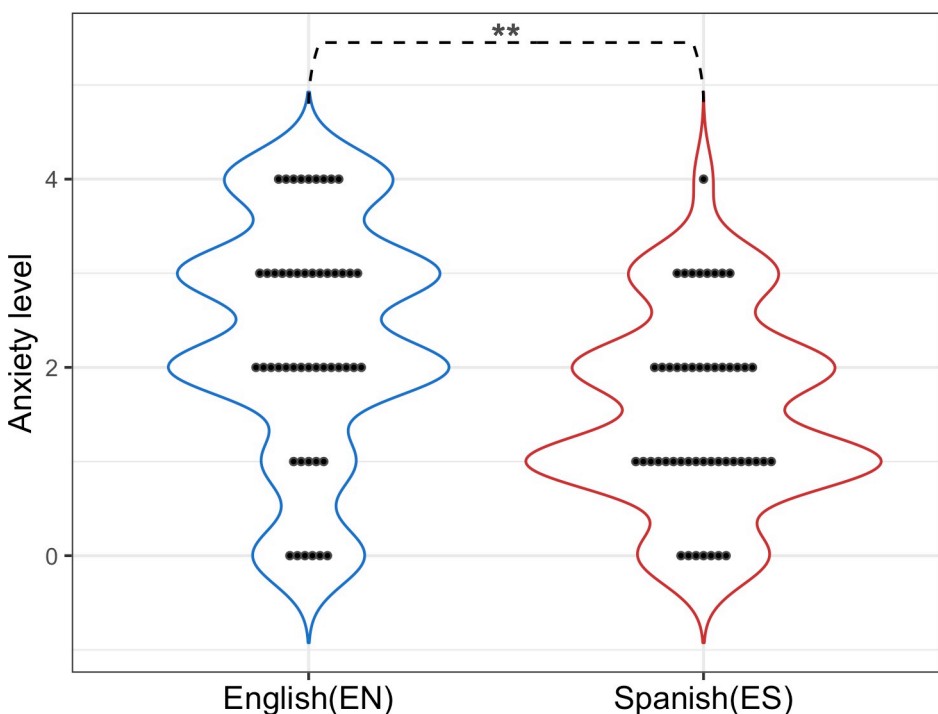

**Fig 5. Anxiety level when making oral presentations in English (EN) vs. Spanish (ES).** A *Poisson* regression was used to analyze discrete ordinal variants (Anxiety level from 1 to 5). A Chi-square test was carried out (z-value = 8,882, Pr (Chi) = 0.005419 **).

international journals, and therefore in English [37]. A scientist's preference for reading and writing in English could also be due to the prevalence of English as the source for scientific words and phrases, as well as the scientist's need to improve their own English in order to overcome these other barriers [41]. The preference of writing directly in English and not translating may be related to the higher cost of translation in comparison with the revision service (Fig 2). Additionally, scientists are more likely to request a favor for English editing than for a translation [37]. Strong feelings of insecurity or an "inferiority complex" generated by scientific writing in English is one of the most important segregation factors mentioned by EFL speaking researchers and increase the need of constant editing or correction [8, 10, 42]. This difficulty or insecurity is augmented in the introduction and discussion sections of an article [12, 43–46]. However, the "materials and methods" section in an article and understanding scientific terminology are equally understood and used in both languages by the respondents, possibly because most words and expressions in modern science are coined in English [47].

In this study, 43.5% of surveyed researchers reported suffering from rejection or revisions because of aspects related to grammar or style in English writing. Coates [48] shows that there is a greater probability of manuscript rejection by a journal if there are grammatical errors, but Lindsey and Crusan [49] found that seems to be the ethnicity of the EFL researchers but not the grammar that is influencing the text evaluation. Some critical voices disagree with the reviewers' bias hypothesis [50]. This subject is still under controversy, and in this paper, without comparing this trend with native speakers, it is not possible to conclude that rejection because of English writing is worse for EFL researchers. To start to unravel this bias hypothesis, it will be necessary to gather primary data about correlations between the quality of the article and impressions from reviewers on the writing of EFL researchers (with and without ethnicity information). Nevertheless, understanding reviewer comments is more difficult for a

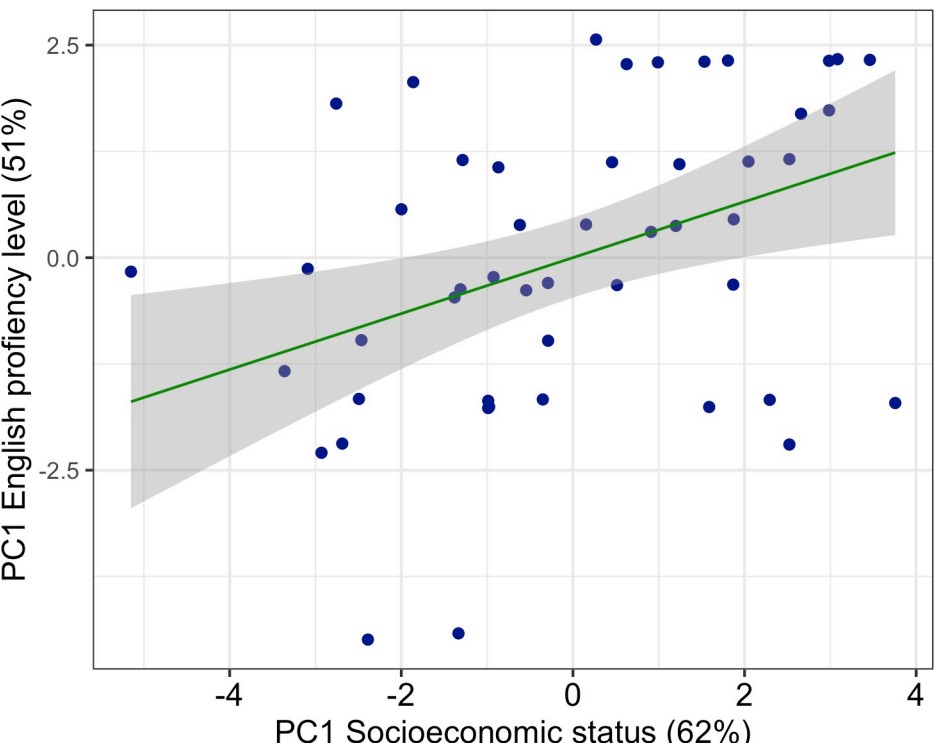

**Fig 6. Relationship between socioeconomic status and English proficiency.** Principle components representing socioeconomic status and English proficiency are significantly correlated ($R^2 = 0.1548$, adjusted $R^2 = 0.1368$, $F = 8.605$, p-value = 0.005168 **).

EFL speaking author, since these frequently contain expressions, euphemisms, or colloquialisms that are not easily interpreted by EFL speakers [51, 52]. For this reason, several authors call on reviewers to write comments that contribute and guide the use of English, and that does not discourage or criticize EFL authors for the lack of mastery of the language [39, 42, 53]. On the other hand, "not every native English speaker is competent to solve peculiarities in the grammar and style of the "good" use of academic English", therefore, all scientists have been pressured to use editing services [54]. In other words, it is questionable to judge or reject innovations or scientific research by linguistic factors or with the excuse of linguistic factors. If a particular research is important for the scientific community, the journal or other resources must assume the cost and effort of translation or editing services, shifting the costs from individual scientists to the publishers or the community.

It was expected that additional costs for Colombian researchers would be found, since similar findings have been reported from other EFL speaking countries in the world [11, 12, 37, 43, 55, 56]. Despite the lack of specific studies on this subject across Latin America, a few exceptions showed similar results: "Regression analysis established that variables of science writing burden contribute to a sense that English is a barrier to scientific writing" [11, 12]. Additionally, opinion pieces from Latin-American researchers also agree about the linguistic barrier in science [39, 57]. It is possible to assume that these results can be extrapolated to other countries bordering Colombia, given the similarity in proficiency and access to English, shared first language, low state investment in science and technology, and parallel political history with the US and Europe [11, 58, 59]. The results could even be extrapolated to other peripheral countries of the world, as Hanauer et al. [12] found similar disadvantages over doctoral students from two countries on different continents, Mexico, and Taiwan.

In this study we not only explore the impact that English proficiency has on doctoral students or post-doctoral researchers, but how those impacts are influenced by the researcher's socioeconomic origin. A positive relationship ($R^2 = 0.14$) was found between English proficiency and socioeconomic status, which is supported by previous studies [60], hence maintaining in science the patterns of social segregation at national and global levels. This low correlation could be explained by a pre-existing socioeconomic bias in Colombia where most undergrad students come from middle and high socioeconomic classes [25, 26]. Another fact that could affect this percentage is the PCA analysis because English proficiency was calculated taking into account years living in English-speaking countries and the percentage of English spoken every day. Therefore, if the researcher lives outside Colombia and speaks English every day the score is higher.

This low correlation, could be explained by the pre-existing socioeconomically biased in Colombia where most undergrad students come from middle and high socioeconomic classes [25, 26]. Another fact that could affect this percentage is the PCA analysis because the English proficiency was calculated taking into account years living in English-speaking countries and the percentage of English spoke every day. Therefore, if the researcher lives outside Colombia and speaks English every day the score is higher.

This study finds that the system within science that denotes English as the lingua franca reinforces inequities between scientists from NES and EFL speaking countries, as well as socioeconomic inequities within countries that primarily speak a language other than English. Globalizing science, so far, has meant offering greater advantages to English speakers at the expense of another scientists' prosperity in the world. Science at present, due to different pressures, opts for English as the only language acceptable for scientific communication, however, some researchers still value the protection of multilingualism in science [44, 61]. Defending multilingualism as an alternative in science would promote the reduction of international and social inequities, which would ultimately boost what Segatto [62] has called "a radically plural world". The homogenization of language in science with the excuse of "integration" is an expression of the elimination of diversity, and this can have consequences not only on the human diversity that makes science but on the diversity of scientific questions that arise [17].

The convenience of a common language in science must be recognized; however, it is essential that solutions to this problem involve scientists from a variety of backgrounds through a bilateral effort (EFL speaking scientists and NES speaking scientists) [10, 16]. Although research is a collective process, the proposed solutions so far have leaned on individual investment, which creates barriers to performing science that more greatly affect researchers of lower socioeconomic backgrounds. Universities, publishers, translation technology, conferences, among others, must also commit to generate ideas for change [17, 37]. One potential approach would be to increase the perceived value of publishing in regional or smaller journals regardless of impact factors (IFs), in order to reduce the pressure to publish in the most prestigious and monolingual journals [6, 63]. Publishing in high IFs journals is a symbolic capital that delineates what should be "desired" as the maximum "goal" of any scientist. In terms of self-identification, not being able to publish in these journals increases the feeling of incompetence and insignificancy [64]. The value given to these IFs journals is supported by the idea that the most important and novel studies in academia are published there, however, an increasing number of voices have highlighted the relative value of scientific advances. For example, differential importance between countries or local communities [18], the influence of trends and use of novel technologies in determining research value (e.g. genetic or genomic data) [65], and devaluation of important but not modern topics in biology, such as natural history and taxonomy [66–68]. Implementing changes in this regard must be a collective effort as we need to rethink the value of scientific publishing. Elife journal is one example of

reevaluating standards in a scientific journal [64]. Other ideas such as encouraging researchers either from the global south or global north who work in the global south to publish in local journals, could be also implemented.

Other alternatives include supporting journals that accept papers in several languages, promoting the inclusion of other languages in journals at the international level, incorporating revision or translation services in all fees paid to publish an article and providing these services to all scientists at no additional charge to them, establishing multilingual annual or periodic editions in renowned journals, among others [37, 57]. Proposals for universities and conferences include aids such as English tutoring for academic purposes [69], retaining in international conferences a space for presenting in local languages [17], using methodologies such as simultaneous translation in conferences, and generating exchange spaces in other languages, among others. Finally, it would be helpful to strengthen public available technologies such as Google Translate that allow simultaneous written translation [17]. In the future, more alternatives will arise, and it will be essential to analyze and monitor them to investigate their reception at the editorial and scientific level.

## Supporting information

**S1 File. Complete article in Spanish.**
(PDF)

**S2 File. Survey questions in Spanish.** Questions in Spanish (original language) of the survey "Implications of language in scientific publications".
(PDF)

**S3 File. Survey questions in English.** Questions in English of the survey "Implications of language in scientific publications".
(XLSX)

**S4 File. Raw data.** Raw data obtained from the Survey in Spanish (original language).
(XLSX)

**S5 File. Theorical framework.** Short explanation of English as lingua franca in Science, English as a foreign language in Colombia and Implication of English in Science (in English and Spanish).
(PDF)

## Acknowledgments

Thanks to the researchers who completed the surveys or helped to share the survey. To Maria Carme Junyent Figueras for being the master thesis director that leads to this paper. To Pere Francesch Rom, Henry Arenas, Prof. Francesc Bernat, Prof. David Bueno and Prof. Avel·lí for editing and making suggestions on the original manuscript in Spanish. To the developers of Google Translate for creating a free powerful tool to translate in the first place the manuscript. To Rebecca Tarvin, Danny Jackson and Tyler Douglas and for editing and commenting on the manuscript in English.

## Author Contributions

**Conceptualization:** Valeria Ramírez-Castañeda.

**Data curation:** Valeria Ramírez-Castañeda.

**Formal analysis:** Valeria Ramírez-Castañeda.

**Investigation:** Valeria Ramírez-Castañeda.

**Methodology:** Valeria Ramírez-Castañeda.

**Project administration:** Valeria Ramírez-Castañeda.

**Resources:** Valeria Ramírez-Castañeda.

**Supervision:** Valeria Ramírez-Castañeda.

**Validation:** Valeria Ramírez-Castañeda.

**Visualization:** Valeria Ramírez-Castañeda.

**Writing – original draft:** Valeria Ramírez-Castañeda.

**Writing – review & editing:** Valeria Ramírez-Castañeda.

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
