## [Decision Letter · Decision Letter 0]

29 Apr 2020

PONE-D-20-06485

Disadvantages of writing, reading, publishing and presenting scientific papers caused by the dominance of the English language in science: The case of Colombian Ph.D. in biological sciences

PLOS ONE

Dear Ramirez-Castaneda,

Thank you for submitting your manuscript to PLOS ONE. After careful consideration, we feel that it has merit but does not fully meet PLOS ONE’s publication criteria as it currently stands. Therefore, we invite you to submit a revised version of the manuscript that systematically addresses all the points raised by the three reviewers during the review process (see below).

We would appreciate receiving your revised manuscript by Jun 13 2020 11:59PM. To enhance the reproducibility of your results, we recommend that if applicable you deposit your laboratory protocols in protocols.io, where a protocol can be assigned its own identifier (DOI) such that it can be cited independently in the future. For instructions see: http://journals.plos.org/plosone/s/submission-guidelines#loc-laboratory-protocols

We look forward to receiving your revised manuscript.

Kind regards,

Emmanuel Manalo, PhD

Academic Editor

PLOS ONE

Journal Requirements:

2. Thank you for including a copy of your questionnaire in Spanish. It is not under a copyright more restrictive than CC-BY, please also include a copy in English, as Supporting Information

4.  We note that Figure 7 in your submission contain copyrighted images. All PLOS content is published under the Creative Commons Attribution License (CC BY 4.0), which means that the manuscript, images, and Supporting Information files will be freely available online, and any third party is permitted to access, download, copy, distribute, and use these materials in any way, even commercially, with proper attribution. For more information, see our copyright guidelines: http://journals.plos.org/plosone/s/licenses-and-copyright.

a)    You may seek permission from the original copyright holder of Figure(s) [#] to publish the content specifically under the CC BY 4.0 license.

Additional Editor Comments (if provided):

The three reviewers of your paper agree that the topic you deal with is very important and you can make a valuable contribution through the report of the research you have undertaken. However, all of them also indicate modifications that are necessary for the paper to reach a standard suitable for publication. These modifications concern contextualisation of the study in the relevant research literature, better support and/or more appropriate expression of some of the claims you make, and clarification of expressions, methods used and data collected. Please go through their comments and suggestions very carefully and address all of the issues they raise in a systematic manner.

Reviewers' comments:

Reviewer's Responses to Questions

**Comments to the Author**

1. Is the manuscript technically sound, and do the data support the conclusions?

Reviewer #1: Partly

Reviewer #2: Partly

Reviewer #3: Partly

2. Has the statistical analysis been performed appropriately and rigorously? 

Reviewer #1: I Don't Know

Reviewer #2: Yes

Reviewer #3: Yes

3. Have the authors made all data underlying the findings in their manuscript fully available?

Reviewer #1: Yes

Reviewer #2: No

Reviewer #3: Yes

4. Is the manuscript presented in an intelligible fashion and written in standard English?

Reviewer #1: Yes

Reviewer #2: No

Reviewer #3: Yes

5. Review Comments to the Author

Reviewer #1: This is an interesting article with a great deal of potential but I think the authors need to consider some of the claims they are making a little more carefully. The title needs rethinking. It could be shorter and clearer. I was also puzzled by the phrase Colombian PhD. It appears the authors are referring to academics who have just completed their PhDs or are enrolled in doctoral studies in biological sciences and are attempting to publish. This should be spelt out clearly in the article and we need more information about the participants.

It would have been a good idea to attach the survey used in the research as it is difficult to judge how appropriate and useful questions are from a summary. I did not find Table 1 particularly useful, in fact, in places it was very confusing. The problem with this kind of survey is that very often the devil is in the detail. For example, some articles are more difficult to write than others simply because of the nature of the article. However, I accept that writing in a second language is far more time-consuming.

The authors claim that English translating and editing services are expensive and I accept this. However, on p.9 it is said that "43.5% Of the doctoral students stated at least one rejection or revision of their articles because of the English grammar". The literature shows that a number of L2 authors believe that their articles are rejected because of their English grammar. However, the evidence to back this up is not convincing, and many editors (and reviewers) contest this strongly. It appears likely that there might well be unconscious bias at least on the part of L1 reviewers and editors, but there is also a fair amount of evidence in the literature to indicate that some of the articles might simply not be up to scratch. The authors revisit this point on p.14. I suggest they read more widely in the area and hedge their claims in this regard. There is simply not enough evidence to back up what they are saying.

The authors make some good points in the discussion. One of the points they raised is the cost of "favours". This is an interesting point but they need to expand on what they mean by the social cost of favours.

On p.16 the authors propose reducing the perceived value of publishing in journals with high impact factors. I would like this to be thought through a little more clearly. If the value of publishing in these journals is reduced then their influence is also reduced. If they are publishing high-quality research is this really in the best interests of the Academy? Also, how could/would this be implemented? The idea of including multiple languages in journals is also an interesting one but this also needs to be thought about in greater detail. It could very easily turn into a politically correct exercise where a few odd articles not written in English are simply inserted. The authors also need to take into account that the most prestigious journals do not charge for publication so where would the money come from to pay for translation services? The journals would need to look to the publishing houses.

Reviewer #2: I think this is a great article investigating a critically important issue in academia – language barriers to early career researches in the Global South. I enjoyed reading it and only have some minor comments, which you can find below.

L51: Johnson et al 2018 – is this an appropriate reference here?

L64: number of researches – should this be the number of researchers?

L69: Note that there is a similar language-driven inequality even within countries in the Global North.

L87: this kind of knowledge – what kind of knowledge? Please elaborate this a bit more.

L86-91: This sentence is very long and I would suggest separating it into two or three.

L94: expatriation – I am not sure if this is a right word. How about “brain drain”?

L101: I don’t think this is an issue only for doctoral students, so the description should be generalised like “… with the socioeconomic origin of researchers”.

L115: This could be my biggest and only concern on this study – how were these 49 doctoral students selected? This kind of surveys needs to avoid any biases (e.g., in gender, socioeconomic status, or any other factors potentially affecting results) in samples as much as possible. The authors need to explain the selection process in more detail and also provide justification that the results did not suffer from effects of those potential biases in samples.

L203-205: This part was a little unclear. How about “To analyze the difficulty of writing scientific articles in two languages, survey participants were also asked how they found it difficult to write different sections of articles: introduction, methods, results, discussion and conclusions.”?

L247-249: The author needs to explain a little more about why these results can be extrapolated to all South American researchers.

L270-273: Little unclear. Please rephrase this part.

L283: I couldn’t understand which part of Fig 2 shows “the cost of translation service that is three times the service for revision”. According to Fig2, translation costs ~500 while standard editing costs ~ 270.

L296: Also see this relevant article: https://doi.org/10.1016/j.tree.2010.07.001

Figs 1 and 2: Avoid using the red-green combination for colour-blind friendliness.

Fig. 2: Y-axis should start from 0 so that we can compare these costs with the average PhD salary (green line) properly.

Fig. 4: I would suggest reordering these so that the introduction is on the left, followed by methods, results, discussion and conclusion towards right.

The journal seems to require authors to make all data fully available but I couldn’t find the data on the survey result.

Reviewer #3: Being a native speaker of Spanish myself, I find this type of articles particularly interesting for personal reasons. I totally agree with the author in the fact that there is a not so small linguistic barrier when it comes to scientific publications for those of us who are not native speakers of English. On a personal note, I must confess I was about to write this review in Spanish, just as a proof of both the need and the usefulness of using such language in academic environments.

Having said this, I will highlight the ‘global English accent’ I can perceive while reading the paper. That is the reason why I will not point out minor mistakes, or maybe I should call them deviations from the standard, that appear in the author’s written expression. Global English is definitely an issue to be mentioned and, more importantly, a theoretical aspect that needs to be addressed deeper than it is in its current version. The use of global English nowadays and the fact that most English speakers worldwide are not native is another reason why proofreading becomes controversial: according to which English standards is the paper proofread? Why should American or British standards be considered higher level than global English, and where is the border line among them?

Although my suggestion to the Editor is to publish this paper with some revisions, there is still a major issue concerning the number of participants in the study. 49 participants is, for a quantitative study using questionnaires, a rather small sample with very little statistical impact. I find it surprising that only 15% of the English proficiency is explained by socioeconomic factors; maybe these data is consequence of the small sample.

In any case, and if the finding of new participants is not feasible at this point of the study, there is something that really needs to be included (as I haven’t seen it as such), and that is a theoretical framework. Following my recommendations aforementioned, I strongly recommend the author to include a section before the Methodology or in the Discussion in which she addresses the most relevant aspects of global English. Global English is the reason why this paper should be published and it seems to have been ignored in the process.

6. PLOS authors have the option to publish the peer review history of their article (what does this mean?). If published, this will include your full peer review and any attached files.

Reviewer #1: No

Reviewer #2: No

Reviewer #3: No

---

## [Author Response · Author response to Decision Letter 0]

17 Jun 2020

Manuscript PONE-D-20-06485

Response to reviewers 

Dear Dr. Emmanuel Manalo,

I sincerely appreciate the opportunity to resubmit a revised draft of the manuscript PONE-D-20-06485 for publication in the PLOS ONE journal. Additionally, I would like to thank the journal and reviewers for your valuable feedback and comments. I have incorporated most of the suggestions made by the reviewers. Changes are tracked in the revised manuscript. Please see below, in blue, the response to the journal requirements and reviewers’ comments and concerns. 

Journal Requirements:

Author response: I sincerely apologize for the lack of attention in the style requirements. I have made several changes in the file naming, reference style, headings, among others. I hope these changes address all the requirements. 

2. Thank you for including a copy of your questionnaire in Spanish. It is not under a copyright more restrictive than CC-BY, please also include a copy in English, as Supporting Information

Author response: Thank you, the survey in English was attached in the Supplementary Information. 

Author response: Sorry for the inconvenience. I have now attached the survey’s raw data as supporting information following the General guidelines for human research participant data (location and birth date were deleted). The raw data was attached in the original language of the survey. Please let me know if you need a translation of the raw data. 

4. We note that Figure 7 in your submission contains copyrighted images. All PLOS content is published under the Creative Commons Attribution License (CC BY 4.0), which means that the manuscript, images, and Supporting Information files will be freely available online, and any third party is permitted to access, download, copy, distribute, and use these materials in any way, even commercially, with proper attribution. For more information, see our copyright guidelines: http://journals.plos.org/plosone/s/licenses-and-copyright.

Author response: This figure was removed from the manuscript.

Reviewer #1

This is an interesting article with a great deal of potential but I think the authors need to consider some of the claims they are making a little more carefully.

Author response: Thank you! I hope I addressed your comments adequately. 

The title needs rethinking. It could be shorter and clearer. 

Author response: Thanks for pointing this out. To address your comment, I have changed the long title from “Disadvantages of writing, reading, publishing and presenting scientific papers caused by the dominance of the English language in science: The case of Colombian Ph.D. in biological sciences” to “Disadvantages in preparing and publishing scientific papers caused by the dominance of the English language in science: The case of Colombian researchers in biological sciences”. The last title includes the fact that the participants are either Ph.D. students or researchers with a doctoral degree. 

 The short title was also changed from “Consequences of English linguistic hegemony in science: The case of Colombian Ph.D. in biological sciences” to “Consequences of English linguistic hegemony in science: The case of Colombian biologist”.

I was also puzzled by the phrase Colombian PhD. It appears the authors are referring to academics who have just completed their PhDs or are enrolled in doctoral studies in biological sciences and are attempting to publish. This should be spelt out clearly in the article and we need more information about the participants.

Author response: I tried to clarified this by adding the following phrase at the beginning of the Methods section: “In order to determine the costs of publishing in English experienced by Colombian researchers in biological sciences 49 to academics were surveyed. These researchers completed their PhDs or are enrolled in doctoral studies and are attempting to publish.”

It would have been a good idea to attach the survey used in the research as it is difficult to judge how appropriate and useful questions are from a summary. 

Author response: Thank you for the comment. You are right and the raw data from the survey has now been attached as supporting information in the original language (Spanish). Birthdate and location were deleted to follow the general guidelines for human research participant data.

I did not find Table 1 particularly useful, in fact, in places it was very confusing. 

Author response: This table was attached to explain how the survey was organized to perform the statistical analysis which is especially crucial when performing the Principal Components Analysis. I opted to maintain table 1. However, I changed the name from “Table 1. Summary of survey questions divided by groups used for correspondent statistical analysis” to “Table 1. Grouping of survey questions in statistical analyses.”. This new title highlights that the table is explaining the groups used for statistical analysis instead of a summary of the survey questions. 

The problem with this kind of survey is that very often the devil is in the detail. For example, some articles are more difficult to write than others simply because of the nature of the article. However, I accept that writing in a second language is far more time-consuming.

Author response: I agree that surveys imply a lot of subjectivity, I tried to use the most quantitative approach as possible. However, I cannot deny there is a human interpretation in each question. This issue was addressed in the methods section: “This survey sought for the most quantitative approach as possible, however, each question is inevitably under some degree of subjectivity due to human interpretation.”

The authors claim that English translating and editing services are expensive and I accept this. However, on p.9 it is said that "43.5% Of the doctoral students stated at least one rejection or revision of their articles because of the English grammar". The literature shows that a number of L2 authors believe that their articles are rejected because of their English grammar. However, the evidence to back this up is not convincing, and many editors (and reviewers) contest this strongly. It appears likely that there might well be unconscious bias at least on the part of L1 reviewers and editors, but there is also a fair amount of evidence in the literature to indicate that some of the articles might simply not be up to scratch. The authors revisit this point on p.14. I suggest they read more widely in the area and hedge their claims in this regard. There is simply not enough evidence to back up what they are saying.

Author response: I agree that there is not enough evidence, especially because there are just a few articles that actually correlate the quality of the article and impressions from reviewers on the writing (with and without ethnicity information). I found one article that tries to do something similar and they found that seems to be the ethnicity but not the grammar that is influencing the bias “Research has repeatedly shown that discrepancies continue to exist between assessments of NES and NNES writers. Our study focuses on determining whether identifying a writer as an international student contributes to such discrepancies. Our survey results suggest they do, and that such discrepancies may be due solely to the perceived ethnolinguistic identity of the writer rather than because of any measurable differences in the writing itself”(Lindsey & Crusan, 2011). Everything outside this is just people's perception and experiences. However, self-perception is important for encouraging or demoralizing EFL researchers (Romero-Olivares, 2019). The most critical voice about the bias hypothesis is Hyland’s (2016) paper, but I think that some of his ideas lack primary data support in the relationship between quality-ethnicity-grammar. 

In the text, this paragraph was changed to “In this study, 43.5% of surveyed researchers reported suffering from rejection or revisions because of aspects related to grammar or style in English writing. Coates [48] shows that there is a greater probability of manuscript rejection by a journal if there are grammatical errors, but Lindsey and Crusan [49] found that it seems to be the ethnicity of the EFL researchers but not the grammar that is influencing the text evaluation. Some critical voices disagree with the reviewers’ bias hypothesis [50]. This subject is still under controversy, and in this paper, without comparing this trend with native speakers, it is not possible to conclude that rejection because of English writing is worse for EFL researchers. To start to unravel this bias hypothesis, it will be necessary to gather primary data about correlations between the quality of the article and impressions from reviewers on the writing of EFL researchers (with and without ethnicity information).”

The authors make some good points in the discussion. One of the points they raised is the cost of "favours". This is an interesting point but they need to expand on what they mean by the social cost of favours.

Author response: Thank you for your comment. I added more information in the manuscript: “More than 90% of researchers have asked for English-editing favors, but favors are unpaid labor that may have a subsequent cost. The cost of this favor particularly leans on the weakest in the relationship, in this case, the EFL researchers because their career depends on publishing in a second language. Therefore, ensuring a permanent source of “favors” is essential for an EFL researcher that is willing to negotiate for “help” by reinforcing dependence with research groups or scientists in NES countries [8]. Romero-Olivares [39] exemplified this point by showing a reviewer comment “The authors need a native English-speaking co-author to thoroughly revise the grammar of this manuscript.”, or as Ordoñez-Matamoros et al [40] show for Colombian researchers “co-authoring with partners located in foreign countries tend to publish their work in journals of higher impact factor and receive more citations per article than those not co-authoring with partners located overseas”.

On p.16 the authors propose reducing the perceived value of publishing in journals with high impact factors. I would like this to be thought through a little more clearly. If the value of publishing in these journals is reduced then their influence is also reduced. If they are publishing high-quality research is this really in the best interests of the Academy? Also, how could/would this be implemented?

Author response: I agree that my intentions were not to devalue IFs of journals but increasing the value of other journals, therefore, I reframed and complemented this statement: 

“One potential approach would be to increase the perceived value of publishing in regional or smaller journals regardless of impact factors (IFs), in order to reduce the pressure to publish in the most prestigious and monolingual journals [6,63]. Publishing in high IFs journals is a symbolic capital that delineates what should be “desired” as the maximum “goal” of any scientist. In terms of self-identification, not being able to publish in these journals increases the feeling of incompetence and insignificancy [64]. The value given to these IFs journals is supported by the idea that the most important and novel studies in academia are published there, however, an increasing number of voices have highlighted the relative value of scientific advances. For example, differential importance between countries or local communities [18], the influence of trends and use of novel technologies in determining research value (eg. genetic or genomic data) [65], and devaluation of important but not modern topics in biology, such as natural history and taxonomy [66–68]. Implementing changes in this regard must be a collective effort as we need to rethink the value of scientific publishing. Elife journal is one example of reevaluating standards in a scientific journal [64]. Other ideas such as encouraging researchers either from the global south or global north who work in the global south to publish in local journals, could be also implemented.”

 The idea of including multiple languages in journals is also an interesting one but this also needs to be thought about in greater detail. It could very easily turn into a politically correct exercise where a few odd articles not written in English are simply inserted. 

Author response: You are right, we need to think about this in more detail, taking into account costs, journals, languages to include, among others. Solutions and resources should increase while visualizing the problem. The last paragraph is dedicated to listing some ideas: “Other alternatives include supporting journals that accept papers in several languages, promoting the inclusion of other languages in journals at the international level, incorporating revision or translation services in all fees paid to publish an article and providing these services to all scientists at no additional charge to them, establishing multilingual annual or periodic editions in renowned journals, among others [37,57]. Proposals for universities and conferences include aids such as English tutoring for academic purposes [69], retaining in international conferences a space for presenting in local languages [17], using methodologies such as simultaneous translation in conferences, and generating exchange spaces in other languages, among others. Finally, it would be helpful to strengthen public available technologies such as Google Translate that allow simultaneous written translation [17]. In the future, more alternatives will arise, and it will be essential to analyze and monitor them to investigate their reception at the editorial and scientific level.”

The authors also need to take into account that the most prestigious journals do not charge for publication so where would the money come from to pay for translation services? The journals would need to look to the publishing houses.

Author response: While I may not know the most prestigious journals in the reviewer’s field, in biology, Nature, PNAS and Science are some of the journals with highest IFs, and all of them charge for publication. About the cost of translating, there are a lot of possibilities. As other difficulties that we have overcome, I am sure that if we understand the importance of increasing diversity in science, we (journals, universities, institutes, governments) could build connections with translators' services in an affordable way. 

Reviewer #2: 

I think this is a great article investigating a critically important issue in academia – language barriers to early career researchers in the Global South. I enjoyed reading it and only have some minor comments, which you can find below.

Author response: Thank you very much!

L51: Johnson et al 2018 – is this an appropriate reference here?

Author response: I agree that a different reference would be more appropriate. I changed this reference for Gordin's 2015 book where the history of language in science is fully explained (L53). Gordin MD. Scientific Babel. Scientific Babel. 2015. doi:10.7208/chicago/9780226000329.001.0001

L64: number of researches – should this be the number of researchers?

Author response: I have corrected this typo on L66.

L69: Note that there is a similar language-driven inequality even within countries in the Global North.

Author response: Absolutely right, therefore, I mention that most of the inequalities are driven by high English proficiency and I listed the countries that I referred as the global north: “Therefore, the prevalence of the English language in the sciences deepens the inequality in knowledge production between countries with high and low English proficiency [5], maintaining the gap in scientific production between the countries of the global south or peripheral and the countries of the global north (include the G8 countries and Australia)...”

L87: this kind of knowledge – what kind of knowledge? Please elaborate this a bit more.

Author response: I complete this sentence by adding: “Therefore, readers with language barriers only have access to limited studies that the researchers consider not complete, important or broad enough to be published in an international journal.”.

L86-91: This sentence is very long and I would suggest separating it into two or three.

Author response: Thanks for pointing this out. I divided this paragraph into two and deleted redundant words: “Therefore, readers with language barriers only have access to limited studies that the researchers consider not complete, important or broad enough to be published in an international journal. Local readers often are unaware of the most significant research being conducted in their region, which has resulted in a void in information important for political decision making, environmental policies, and conservation strategies [16–18].”.

L94: expatriation – I am not sure if this is a right word. How about “brain drain”?

Author response: I have added the suggested content to the manuscript on L96.

L101: I don’t think this is an issue only for doctoral students, so the description should be generalised like “… with the socioeconomic origin of researchers”.

Author response: I have added the suggested content to the manuscript on L103.

L115: This could be my biggest and only concern on this study – how were these 49 doctoral students selected? This kind of survey needs to avoid any biases (e.g., in gender, socioeconomic status, or any other factors potentially affecting results) in samples as much as possible. The authors need to explain the selection process in more detail and also provide justification that the results did not suffer from effects of those potential biases in samples.

Author response: I agree that this is a potential limitation of the study. The survey was available online for two months and it was shared personally to researchers (who also reshared the survey) and on twitter under the hashtag “CienciaCriolla” that is used between Colombian researchers. It was also anonymous, therefore, I did not select who answered them or controlled by any bias. However, controlling this data by bias is very difficult because, in Colombia, becoming a researcher is already biased. Only 30.21% of natural science researchers are women, most of the researchers come from the biggest cities in Colombia, and most of undergrad students come from middle and high socioeconomic classes.

I added this information in the Methods section: “This survey was available for two months and shared directly to researchers and on Twitter under the hashtag “#CienciaCriolla” (used between Colombian researchers). Responses were anonymous. It must be mentioned that the researcher’s demography in Colombia is gender, ethnic, and socioeconomic biased. Only 30.21% of natural science researchers are women [24], researchers come primarily from big cities [25], and undergrad students come mainly from middle and high socioeconomic classes [26]. Therefore, it would not have been possible to completely control for bias in who took the survey.”

L203-205: This part was a little unclear. How about “To analyze the difficulty of writing scientific articles in two languages, survey participants were also asked how they found it difficult to write different sections of articles: introduction, methods, results, discussion, and conclusions.”?

Author response: Thank you very much for your suggestion, it was used in place of “To analyze the difficulty of writing scientific articles in two languages, survey participants were also asked how they found it difficult to write different sections of articles: introduction, methods, results, discussion, and conclusions”.

L247-249: The author needs to explain a little more about why these results can be extrapolated to all South American researchers.

Author response: Thanks, this statement was explained later in the discussion section. However, I added additional information: “Despite the lack of specific studies on this subject across Latin America, a few exceptions showed similar results: “Regression analysis established that variables of science writing burden contribute to a sense that English is a barrier to scientific writing” [11,12]. Additionally, opinion pieces from Latin-American researchers also agree about the linguistic barrier in science [39,57]. It is possible to assume that these results can be extrapolated to other countries bordering Colombia, given the similarity in proficiency and access to English, shared first language, low state investment in science and technology, and parallel political history with the US and Europe [11,58,59]. The results could even be extrapolated to other peripheral countries of the world, as Hanauer et al. [12] found similar disadvantages over doctoral students from two countries on different continents, Mexico, and Taiwan. ”

L270-273: Little unclear. Please rephrase this part.

Author response: Thank you for your comment. I rephrased it and added more information in the manuscript: “More than 90% of researchers have asked for English-editing favors, but favors are unpaid labor that may have a subsequent cost. A cost particularly leans on the weakest in the relationship, in this case, the EFL researchers because their career depends on publishing in a second language. Therefore, ensuring a permanent source of “favors” is essential for an EFL researcher that is willing to negotiate for “help” by reinforcing dependence with research groups or scientists in NES countries [8]. Romero-Olivares [39] exemplified this point by showing a reviewer comment “The authors need a native English-speaking co-author to thoroughly revise the grammar of this manuscript.”, or as Ordoñez-Matamoros et al [40] show for Colombian researchers “co-authoring with partners located in foreign countries tend to publish their work in journals of higher impact factor and receive more citations per article than those not co-authoring with partners located overseas”.

L283: I couldn’t understand which part of Fig 2 shows “the cost of translation service that is three times the service for revision”. According to Fig2, translation costs ~500 while standard editing costs ~ 270.

Author response: My apologies, you are right. I replaced this phrase with “The preference of writing directly in English and not translating may be related to the higher cost of translation in comparison with the revision service (Fig 2).”.

L296: Also see this relevant article: https://doi.org/10.1016/j.tree.2010.07.001

Author response: Thanks for your suggestion, I cited Clavero’s (2010) article along with Huang (2010), and Romero-Olivares (2019): “For this reason, several authors call on reviewers to write comments that contribute and guide the use of English, and that does not discourage or criticize EFL authors for the lack of mastery of the language [39,42,53]. ”

Figs 1 and 2: Avoid using the red-green combination for colour-blind friendliness.

Author response: Thank you for pointing this out. I changed the green line for a dotted line in both graphs. 

Fig. 2: Y-axis should start from 0 so that we can compare these costs with the average PhD salary (green line) properly.

Author response: Thanks for pointing this out. I changed the y-axis limits in fig 2.

Fig. 4: I would suggest reordering these so that the introduction is on the left, followed by methods, results, discussion and conclusion towards right.

Author response: I agree and I rearranged the order in figure’s 4 y-axis. 

The journal seems to require authors to make all data fully available but I couldn’t find the data on the survey result.

Author response: Thank you for the comment. You are right and the raw data from the survey is now attached as supporting information in the original language (Spanish). Birthdate and location were deleted to follow the general guidelines for human research participant data.

Reviewer #3: 

Being a native speaker of Spanish myself, I find this type of articles particularly interesting for personal reasons. I totally agree with the author in the fact that there is a not so small linguistic barrier when it comes to scientific publications for those of us who are not native speakers of English. On a personal note, I must confess I was about to write this review in Spanish, just as a proof of both the need and the usefulness of using such language in academic environments.

Author response: Muchas gracias por tu empatía y sinceridad en tu comentario. 

Having said this, I will highlight the ‘global English accent’ I can perceive while reading the paper. That is the reason why I will not point out minor mistakes, or maybe I should call them deviations from the standard, that appear in the author’s written expression. Global English is definitely an issue to be mentioned and, more importantly, a theoretical aspect that needs to be addressed deeper than it is in its current version. The use of global English nowadays and the fact that most English speakers worldwide are not native is another reason why proofreading becomes controversial: according to which English standards is the paper proofread? Why should American or British standards be considered higher level than global English, and where is the border line among them?

Author response: I agree with your comment. 

Although my suggestion to the Editor is to publish this paper with some revisions, there is still a major issue concerning the number of participants in the study. 49 participants is, for a quantitative study using questionnaires, a rather small sample with very little statistical impact. 

Author response: I understand your point. Although I did not find specific numbers for Colombian researchers in biological sciences, according to the UNESCO database in Colombia there are 58 researchers (from all career stages and all disciplines) per million inhabitants (50 million in Colombia). Doing raw calculations: 6 disciplines (social sciences, agricultural sciences, medicine, and health sciences, basic science, engineering, and natural sciences), and 4 different stages (undergrad, masters, Ph.D. student, postdoc), then we could think that less than 400 of researchers are biologists (Ph.D. students and postdoc). If the population is 400 and the sample is 49 researchers then this sample size is around 10% of the population, which is a decent sample size. 

Furthermore, except for research in Spain, other authors have published similar studies with less or similar sample data or/and unspecific to biological sciences. See the following table:

Author, year

Number of researchers surveyed

Discipline

Country

Reference

Pérez-Llantada et al., 2011

10

Several disciplines

Spain

https://doi.org/10.1016/j.esp.2010.05.001

Karimnia, 2013

10

Several disciplines

Iran

https://doi.org/10.1016/j.sbspro.2013.01.137

Huang, 2010

10

Several disciplines

Taiwan

https://doi.org/10.1016/j.jeap.2009.10.001

McGrath, 2014

15

Several social sciences

Sweden

https://doi.org/10.1016/j.jeap.2013.10.008

Murasan et al., 2014

91

Several disciplines

Romania

https://doi.org/10.1016/j.jeap.2013.10.009

Duszak et al., 2008

99

Several disciplines

Polonia

https://doi.org/10.1016/j.jeap.2008.03.001

Moreno et al., 2012

1717

Several disciplines

Spain

http://eprints.rclis.org/29319/

Ferguson et al., 2011

300 (59 biologist)

Natural and basic sciences

Spain

https://doi.org/10.1111/j.1467-971X.2010.01656.x

Hanuaer et al., 2019

Taiwan: 238, Mexico: 148

Several disciplines

Taiwan and Mexico

https://doi.org/10.1177/0741088318804821

Nevertheless, I recognize that overall 49 is a small sample size from which to draw statistical inferences, and I do mention this in (line 135): “It must be also recognized that without specific numbers for total Colombian researchers in biological sciences, 49 may not be a representative sample size from which to draw accurate statistical inferences.”

I find it surprising that only 15% of the English proficiency is explained by socioeconomic factors; maybe this data is a consequence of the small sample.

Author response: Thanks for pointing this out. I definitely agree that it is surprising. I can think about two other reasons for this result: 1) In Colombia, becoming a researcher is already socioeconomically biased because most researchers come from the biggest cities in Colombia, and most undergrad students come from middle and high socioeconomic classes. 2) The PCA statistical analysis for English proficiency took into account years living in English-speaking countries and the percentage of English spoke every day. Therefore, if the researcher lives outside Colombia and speaks English every day the score is higher, even if they come from lower socioeconomic backgrounds. 

I added this information in the paragraph (Line 400): “This low correlation could be explained by a pre-existing socioeconomic bias in Colombia where most undergrad students come from middle and high socioeconomic classes [25,26]. Another fact that could affect this percentage is the PCA analysis because English proficiency was calculated taking into account years living in English-speaking countries and the percentage of English spoken every day. Therefore, if the researcher lives outside Colombia and speaks English every day the score is higher.”

In any case, and if the finding of new participants is not feasible at this point in the study, there is something that really needs to be included (as I haven’t seen it as such), and that is a theoretical framework. Following my recommendations aforementioned, I strongly recommend the author to include a section before the Methodology or in the Discussion in which she addresses the most relevant aspects of global English. Global English is the reason why this paper should be published and it seems to have been ignored in the process.

Author response: I understand your point. However, I believe a research paper needs to be as concise as possible and this research article is already very long. To not overwhelm the readers but address the important point that you mentioned, I added a supporting file S6 Theoretical framework. This file includes three subtitles 1) English as lingua franca in science, 2) English as a foreign language in Colombia, and 3) Implications of English in science.

---

## [Decision Letter · Decision Letter 1]

17 Aug 2020

Disadvantages in preparing and publishing scientific papers caused by the dominance of the English language in science: The case of Colombian researchers in biological sciences

PONE-D-20-06485R1

Dear Dr. Ramirez-Castaneda,

We’re pleased to inform you that your manuscript has been judged scientifically suitable for publication and will be formally accepted for publication once it meets all outstanding technical requirements.

Kind regards,

Emmanuel Manalo, PhD

Academic Editor

PLOS ONE

Additional Editor Comments (optional):

Reviewers' comments:

Reviewer's Responses to Questions

**Comments to the Author**

1. If the authors have adequately addressed your comments raised in a previous round of review and you feel that this manuscript is now acceptable for publication, you may indicate that here to bypass the “Comments to the Author” section, enter your conflict of interest statement in the “Confidential to Editor” section, and submit your "Accept" recommendation.

Reviewer #1: All comments have been addressed

Reviewer #2: All comments have been addressed

2. Is the manuscript technically sound, and do the data support the conclusions?

Reviewer #1: Yes

Reviewer #2: Yes

3. Has the statistical analysis been performed appropriately and rigorously? 

Reviewer #1: I Don't Know

Reviewer #2: Yes

4. Have the authors made all data underlying the findings in their manuscript fully available?

Reviewer #1: Yes

Reviewer #2: Yes

5. Is the manuscript presented in an intelligible fashion and written in standard English?

Reviewer #1: Yes

Reviewer #2: Yes

6. Review Comments to the Author

Reviewer #1: (No Response)

Reviewer #2: Thank you for dealing thoroughly with my comments on the previous version. I have found most responses satisfactory and now have no major concerns.

Regarding the sample size issue, I think that the author’s response to the third comment by Reviewer #3 is very helpful (i.e., estimated number of Colombian ECRs in biology, as well as sample size in earlier similar studies), and wonder if the author wants to provide it in the main text for justifying the sample size in this study.

I again would like to commend the author for completing this important research.

7. PLOS authors have the option to publish the peer review history of their article (what does this mean?). If published, this will include your full peer review and any attached files.

Reviewer #1: **Yes: **Pat Strauss

Reviewer #2: No

---

## [Editor Report · Acceptance letter]

24 Aug 2020

PONE-D-20-06485R1 

Disadvantages in preparing and publishing scientific papers caused by the dominance of the English language in science: The case of Colombian researchers in biological sciences 

Dear Dr. Ramirez-Castaneda:

I'm pleased to inform you that your manuscript has been deemed suitable for publication in PLOS ONE. Congratulations! Your manuscript is now with our production department. 

Kind regards, 

on behalf of

Professor Emmanuel Manalo 

Academic Editor

PLOS ONE